# LightPAFF: A Two-Stage Distillation Framework for Pre-training and Fine-tuning

## Abstract

While pre-training and fine-tuning, e.g., BERT (Devlin et al., 2018), GPT-2 (Radford et al., 2019), have achieved great success in language understanding and generation tasks, the pre-trained models are usually too big for online deployment in terms of both memory cost and inference speed, which hinders them from practical online usage. In this paper, we propose LightPAFF, a Lightweight Pre-training And Fine-tuning Framework that leverages two-stage knowledge distillation to transfer knowledge from a big teacher model to a lightweight student model in both pre-training and fine-tuning stages. In this way the lightweight model can achieve similar accuracy as the big teacher model, but with much fewer parameters and thus faster online inference speed. LightPAFF can support different pre-training methods (such as BERT, GPT-2 and MASS (Song et al., 2019)) and be applied to many downstream tasks. Experiments on three language understanding tasks, three language modeling tasks and three sequence to sequence generation tasks demonstrate that while achieving similar accuracy with the big BERT, GPT-2 and MASS models, LightPAFF reduces the model size by nearly 5x and improves online inference speed by 5x-7x.

## 1 Introduction

Recently, the pre-training and fine-tuning frameworks (Devlin et al., 2018; Radford et al., 2018; 2019; Yang et al., 2019b; Liu et al., 2019; Lample & Conneau, 2019; Song et al., 2019) have achieved great success in NLP by learning universal language representations on large-scale language corpus and transferring knowledge from the pre-trained model to downstream tasks. Based on the carefully designed unsupervised pre-training strategy, e.g., masked language modeling (Devlin et al., 2018), causal language modeling (Radford et al., 2018; 2019), permutation language modeling (Yang et al., 2019b) and masked sequence to sequence modeling (Song et al., 2019), the pre-trained model can extract semantic information, understand meanings, generate sentences, even capture common-sense knowledge from large scale corpus. Some of the most representative pre-trained models are BERT (Devlin et al., 2018), XLNet (Yang et al., 2019b) for language understanding tasks, GPT/GPT-2 (Radford et al., 2018; 2019) for language modeling tasks, and XLM (Lample & Conneau, 2019), MASS (Song et al., 2019) for sequence to sequence based language generation tasks.

However, pre-trained models are usually of huge model parameter size, e.g., BERT has more than 300 million parameters while GPT-2 has more than 1.5 billion parameters, which brings challenges in memory cost and inference latency when deploying the pre-trained models online or on mobile devices. One straightforward solution is to use smaller model for pre-training and fine-tuning, which will cause much accuracy degradation that cannot be acceptable for online usage. Previous works (Yang et al., 2019a; Tang et al., 2019; Sun et al., 2019) have tried knowledge distillation to reduce the number of parameters for BERT. However, they just distill the knowledge of fine-tuned BERT to the task-specific small model, still with big accuracy drop compared with the original fine-tuned BERT model. Meanwhile, most works focus on compressing BERT while few works study on reducing the size of other models like GPT-2, XLNet, XLM, MASS.

Knowledge distillation (Bucilu et al., 2006; Hinton et al., 2015; Yu et al., 2013) is an effective and practical way to transfer the knowledge from a cumbersome teacher model to a lightweight student model. However, unlike the previous works (Yang et al., 2019a; Tang et al., 2019; Sun et al., 2019)

that only leverage the knowledge in the fine-tuned task-specified model, the knowledge in the pre-trained model is also helpful and even more general for language understanding and generation. In this work, we propose LightPAFF, a Lightweight Pre-training And Fine-tuning Framework that incorporates knowledge distillation in both pre-training and fine-tuning stages and transfers knowledge from a pre-trained big model into a smaller model. As a result, LightPAFF can greatly reduce the number of model parameters and thus memory cost and inference latency, without losing much of accuracy.

As a general framework, LightPAFF can be applied on a variety of pre-trained models (e.g., BERT, GPT/GPT-2, XLNet, RoBERTa, XLM, MASS) and downstream tasks (e.g., language understanding/modeling/generation). In this paper, we carefully choose BERT, GPT-2 and MASS as three study cases, considering the model diversity that BERT leverages masked language modeling for pre-training and is for downstream language understanding tasks, GPT-2 leverages causal language modeling for pre-training and is for language modeling and generation tasks, while MASS leverages masked sequence to sequence modeling for pre-training and is for sequence to sequence based language generation tasks. We formulate LightPAFF on the pre-training and fine-tuning tasks of BERT, GPT-2 and MASS uniformly, and conduct experiments on three language understanding tasks (SST-2, QQP, polyphone disambiguation) for BERT, three language modeling tasks (WikiText-2, PTB, WikiText-103) for GPT-2, and three sequence to sequence based language generation tasks (WMT Zh-En, En-De, En-Fr translation) for MASS. Experimental results demonstrate that LightPAFF reduces the model parameter of BERT/GPT-2/MASS by nearly $5\times$ and improves the inference speed by $5\times \sim 7\times$, while achieves similar accuracy with the original BERT/GPT-2/MASS model [1].

## 2 TWO-STAGE DISTILLATION FRAMEWORK

Different from existing knowledge distillation methods for pre-trained models (Yang et al., 2019a; Tang et al., 2019; Sun et al., 2019), which only perform distillation in the fine-tuning stage, we perform distillation not only in the fine-tuning stage but also in the pre-training stage to ensure the accuracy of the final student model. The basic idea of our two-stage distillation is shown in Figure 1, and we call this Light-weight Pre-training And Fine-tuning Framework as LightPAFF for short.

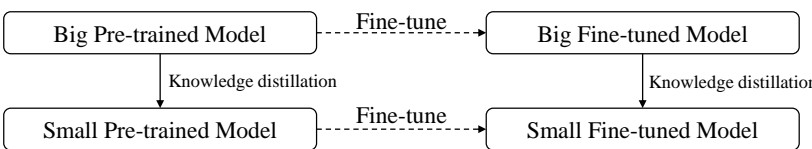

Figure 1: LightPAFF pipeline.

LightPAFF runs in four steps:

1. Pre-train the big teacher model using the pre-training dataset.

2. Fine-tune the big teacher model using the dataset of a downstream task.

3. Distill the pre-trained big teacher model and obtain a light-weight pre-trained model using the pre-training dataset.

4. Fine-tune the light-weight pre-trained model for the downstream task by using the dataset of the task and distilling the fine-tuned big teacher model.

As can be seen, the first and second steps are exactly the same as previous pre-training methods such as BERT (Devlin et al., 2018) and GPT-2 (Radford et al., 2019). The third step performs distillation in the pre-training stage, and the fourth step performs distillation in the fine-tuning stage. We adopt a general formulation for knowledge distillation in LightPAFF:

$$\mathcal{L}(\theta) = \sum_{(x,y)} (1 - \lambda) \cdot MLE(x, y; \theta) + \lambda \cdot KL(P(x; \theta_T), P(x; \theta)), \tag{1}$$

---

[1] We will release the codes and models once accepted.

where $MLE(x, y; \theta)$ is the maximal likelihood loss of the student model $\theta$ over a training data pair $(x, y)$, $KL(P(x; \theta_T), P(x; \theta))$ is the KL divergence between the predicted probability distribution on the data $x$ of the teacher model $\theta_T$ and the distribution of the student model $\theta$, and $\lambda$ is a hyperparameter to trade off the two loss terms.

LightPAFF is a general framework and can be applied to many pre-training fine-tuning tasks. In the following sections, we choose three representative models considering their diversity and code availability: BERT (Devlin et al., 2018), which is popular and effective for language understanding tasks such as GLUE benchmark, GPT-2 (Radford et al., 2019), which is popular and effective for language modeling tasks, and MASS (Song et al., 2019), which is popular and effective for sequence to sequence generation tasks such as machine translation. Detailed descriptions of two-stage knowledge distillation for BERT, GPT-2 and MASS can be found in Appendix (Section 1). We conduct experimental study of LightPAFF on BERT in Section 3, GPT-2 in Section 4 and MASS in Section 5, and then conduct some general analyses of LightPAFF in Section 6.

## 3    LIGHTPAFF FOR BERT

We follow Equation 1 to conduct knowledge distillation for BERT in both pre-training and fine-tuning stages. The BERT teacher model provides the probability distribution of the masked tokens to the student model during pre-training distillation[2], while provides the probability distribution of the class label to the student model during fine-tuning distillation. In the following sections, we describe the experimental setting and results of LightPAFF for BERT.

### 3.1    EXPERIMENTAL SETTING

**Datasets**  For BERT pre-training, we use BooksCorpus (800M words) and Wikipedia (2500M words) for English BERT as used in Devlin et al. (2018), and NewsCorpus (2000M words) and Wikipedia (300M words) for Chinese BERT. For downstream language understanding tasks, we choose SST-2 (Socher et al., 2013) and QQP (Iyer et al., 2017) on English and polyphone disambiguation (PolyDis) (Zhang et al., 2002) on Chinese. Additionally, we add 1M unlabeled data for SST-2, 2M for QQP and 1M for PolyDis in the fine-tuning stage, where we simply use the teacher model to generate probability distribution to train the student model. More descriptions about datasets can be found in Appendix (Section 2).

**Model Configuration**  We denote L, H, A as the number of layers, hidden size and the number of attention heads in Transformer. For BERT teacher model, we choose the pre-trained BERT-base model (L=12, H=768, A=12) with 110M parameters released by Devlin et al. (2018). For BERT student model, we use the configuration (L=3, H=512, A=8) with 25M parameters for English and 20M parameters for Chinese, where the parameter difference is due to the different vocabulary sizes.

**Training and Inference**  The hyperparameter $\lambda$ in Equation 1 is set as 1.0 in pre-training according to the performance of pre-training on a held out validation set, and the $\lambda$ in fine-tuning is chosen according to the validation performance on each downstream task. We pre-train BERT on 8 NVIDIA V100 GPUs, each with 16GB memory, and with batch size of 400 sentences and max sequence length of 128 tokens. We fine-tune on 4 NVIDIA V100 GPUs with batch size of 32 sentences.

### 3.2    RESULTS

**Accuracy of LightPAFF**  The results of LightPAFF on BERT tasks (SST-2, QQP and PolyDis) are shown in Table 1. It can be seen that LightPAFF achieves similar accuracy (maintains nearly 99.5% of accuracy) with the BERT teacher model, but reduces the model size by nearly 5× (from 110M to 25M or 20M). Compared with the model with the same size of parameters (Transformer small and BERT small) and the previous works (Patient KD and Distilled BiLSTM) that only use

---

[2]BERT leverages masked language modeling and next sentence prediction for pre-training. Considering some works (Yang et al., 2019b; Liu et al., 2019; Joshi et al., 2019) have achieved good results without next sentence prediction, we only distill BERT on the masked language modeling task in pre-training.

knowledge distillation in the fine-tuning stage, LightPAFF achieves much better accuracy[3]. The trade-off between accuracy and size of the student model can be found in Appendix (Section 3). Since BERT leverages additional unlabeled data for distillation, we also study the effect of unlabeled data in Appendix (Section 4).

| | SST-2 | | QQP | | PolyDis | |
|---|---|---|---|---|---|---|
| | Acc. | #Param | F1/Acc. | #Param | Acc. | #Param |
| BERT (Devlin et al., 2018) | 93.5 | 110M | 71.2/89.2 | 110M | 95.9 | 110M |
| Patient KD (Sun et al., 2019) | 87.5 | 35M | 68.1/87.8 | 35M | - | - |
| Distilled BiLSTM (Tang et al., 2019) | 90.7 | 10M | 68.2/88.1 | 10M | - | - |
| Transformer small | 83.4 | 25M | 61.1/81.5 | 25M | 89.3 | 20M |
| BERT small | 89.7 | 25M | 65.4/85.1 | 25M | 93.6 | 20M |
| LightPAFF | **92.9** | 25M | **70.6/88.6** | 25M | **95.4** | 20M |

Table 1: Results of LightPAFF on BERT tasks. "BERT" represents the BERT-base model released by Devlin et al. (2018) and is taken as the teacher model, "Patient KD" distills BERT-base model into Transformer model during fine-tuning, "Distilled BiLSTM" distills BERT-large model into LSTM model during fine-tuning, "Transformer small" and "BERT small" share the same model structure and number of parameters, with the only difference that "Transformer small" has no BERT pre-training, "LightPAFF" is our method applied on "BERT small".

**Inference Speedup of LightPAFF** We further measure the inference latency of LightPAFF on both GPU and CPU. As shown in Table 2, LightPAFF achieves about $6\times$ speedup over the big model on GPU and about $7\times$ speedup on CPU in BERT task, demonstrating the advantages of LightPAFF for inference speedup.

| | #Param | Latency (GPU) | Speedup | Latency (CPU) | Speedup |
|---|---|---|---|---|---|
| BERT | 110M | 25 ms | 1.00 $\times$ | 78 ms | 1.00 $\times$ |
| LightPAFF | 25M | 4 ms | 6.25 $\times$ | 11 ms | 7.09 $\times$ |

Table 2: Inference speedup of LightPAFF. Evaluation is conducted on NVIDIA Tesla 16GB V100 GPU or Intel(R) Xeon(R) Platinum 8168 CPU, with batch size of 1 sentence and max sequence length of 128 tokens.

## 4 LIGHTPAFF FOR GPT-2

Since GPT-2 (Radford et al., 2019) uses language modeling for both pre-training and fine-tuning, the knowledge distillation is similar in the two stages, where the GPT-2 teacher model provides the probability distribution of each token in a sentence to the student model during distillation. In the following sections, we describe the experimental setting and results of LightPAFF for GPT-2.

### 4.1 EXPERIMENTAL SETTING

**Datasets** For GPT-2 pre-training, we use the same English training corpus as used in BERT described in last section[4]. For downstream language modeling tasks, we choose WikiText-2 (Merity et al., 2017), PTB (Mikolov et al., 2010) and WikiText103 (Merity et al., 2017). More descriptions about datasets can be found in Appendix (Section 2).

---

[3]Note that Distilled BiLSTM (Tang et al., 2019) uses BERT-large as teacher model while LightPAFF uses BERT-base. LightPAFF can achieve better accuracy by using BERT-large as the teacher model, which is left for future work.

[4]Due to the constraint of computation resource, we do not use the huge training data as in GPT-2 (Radford et al., 2019), but we believe using more data can achieve better accuracy, and leave it for future work.

**Model Configuration** For GPT-2 teacher model, we choose the pre-trained model (L=24, H=1024, A=16, 345M parameters) released by Radford et al. (2019). For student model (we call GPT-2 small), we use the configuration (L=4, H=768, A=12, 67M parameters). L, H, A are same meaning as defined in last Section.

**Training and Inference** The hyperparameter $\lambda$ for pre-training is set as 0.4 according to the zero-shot results, and $\lambda$ for fine-tuning is chosen according to the validation performance on each downstream task. We pre-train on 16 NVIDIA V100 GPU, each with 16GB memory, and with the batch size of 512 sentences and max sequence length of 1024 tokens. We fine-tune on 1 NVIDIA V100 GPU with batch size of 8 sentences and max sequence length of 512 tokens. We report the word-level perplexity via de-tokenizer following Radford et al. (2019).

## 4.2 RESULTS

**Accuracy of LightPAFF** The results of LightPAFF on GPT-2 tasks (WikiText-2, PTB, Wiki-Text103) are shown in Table 3. It can be seen that lightPAFF achieves large improvements than the models under the same size of parameters without pre-training (Transformer small) and with pre-training (GPT-2 small), while approaches the perplexity of the GPT-2 teacher model in the downstream tasks but with only $1/5\times$ number of parameters.

|  | #Param | WikiText-2 | | PTB | | WikiText103 | |
| --- | --- | --- | --- | --- | --- | --- | --- |
|  |  | w/o FT | w/ FT | w/o FT | w/ FT | w/o FT | w/ FT |
| GPT-2 | 345M | 22.8 | 15.5 | 47.3 | 17.0 | 26.4 | 13.0 |
| Transformer small | 67M | - | 71.6 | - | 77.0 | - | 28.5 |
| GPT-2 small | 67M | 58.5 | 25.5 | 129.6 | 31.2 | 61.4 | 19.8 |
| LightPAFF | 67M | **32.6** | **18.8** | **70.0** | **22.8** | **38.7** | **16.4** |

Table 3: Results of LightPAFF on GPT-2 tasks in terms of perplexity (ppl). "w/ FT" and "w/o FT" mean the pre-trained model is evaluated on downstream tasks with and without fine-tuning. "GPT-2" represents the model released by Radford et al. (2019) and is taken as the teacher model, "Transformer small" and "GPT-2 small" share the same model structure and number of parameters, with the only difference that "Transformer small" has no GPT-2 pre-training, "LightPAFF" is our method applied on "GPT-2 small".

**Inference Speedup of LightPAFF** The inference latency of LightPAFF is shown in Table 4. Light-PAFF achieves more than $5\times$ speedup over the big model on GPU while nearly $7\times$ speedup in CPU, which indicates the effectiveness of LightPAFF for inference speedup on GPT-2.

|  | #Param | Latency (GPU) | Speedup | Latency (CPU) | Speedup |
| --- | --- | --- | --- | --- | --- |
| GPT-2 | 345M | 553 ms | $1.00 \times$ | 1683 ms | $1.00 \times$ |
| LightPAFF | 67M | 101 ms | $5.47 \times$ | 245 ms | $6.87 \times$ |

Table 4: Inference speedup of LightPAFF on GPT-2. Evaluation is conducted with batch size of 1 sentence and max sequence length of 512 tokens.The GPU/CPU configurations follow that in last Section.

## 5 LIGHTPAFF FOR MASS

Similar to BERT, the MASS teacher model provides the probability distribution on each token in a masked segment to the student model during pre-training distillation, and provides the probability distribution on each token in a target sequence to the student model during fine-tuning distillation. In the following sections, we describe the experimental setting and results of LightPAFF for MASS.

## 5.1 Experimental Setting

**Datasets** For MASS pre-training, we use 50M monolingual data of each language from the Newscrawl dataset, following Song et al. (2019). For fine-tuning tasks, we choose a rich-resource task (WMT17 Chinese-English, briefly Zh-En) and two low-resource tasks (WMT14 English-French, briefly En-Fr and WMT16 English-German, briefly En-De), where we simulate the low-resource scenario by only choosing a small part of training data. More descriptions about datasets can be found in Appendix (Section 2).

**Model Configuration** For MASS teacher model, we choose the pre-trained model with the configuration (L=6, H=1024, A=16) for both encoder and decoder, which is released by Song et al. (2019). For student model, we use the configuration (L=6, H=512, A=8) for the encoder and (L=4, H=512, A=8) for the decoder. L, H, A are same meaning as described in the BERT section.

**Training and Inference** The hyperparameter $\lambda$ for pre-training is set as 0.7 according to the performance of pre-training on a held out validation set, and $\lambda$ for fine-tuning is adjusted according to the validation performance on each downstream task. We pre-train on 8 NVIDIA V100 GPUs with batch size of 48,000 tokens. We fine-tune on 8 NVIDIA V100 GPUs with batch size of 25,600 tokens. We report the BLEU score by SacreBLEU[5].

## 5.2 Results

**Accuracy of LightPAFF** The results of LightPAFF on MASS tasks are shown in Table 5. It can be seen that compared with the pre-trained model with the same size of parameters (MASS small), LightPAFF achieves 2.5-3.8 BLEU points improvement in low-resource tasks and 1.1 BLEU points improvement in rich-resource task. LightPAFF achieves similar accuracy with the MASS teacher model, but reduces the model size by nearly $5\times$ (from 213M to 42M or from 307M to 67M).

| | WMT17 Zh-En | | WMT16 En-De (100K) | | WMT14 En-Fr (100K) | |
| --- | --- | --- | --- | --- | --- | --- |
| | BLEU | #Param | BLEU | #Param | BLEU | #Param |
| MASS | 25.2 | 307M | 33.1 | 213M | 26.7 | 213M |
| Transformer small | 21.3 | 67M | 23.8 | 42M | 14.1 | 42M |
| MASS small | 23.8 | 67M | 28.4 | 42M | 23.2 | 42M |
| LightPAFF | **24.9** | 67M | **32.2** | 42M | **25.7** | 42M |

Table 5: Results of lightPAFF on sequence to sequence based language generation tasks in terms of BLEU score. "MASS" represents the MASS model released by Song et al. (2019) and is taken as the teacher model, "Transformer small" and "MASS small" share the same model structure and number of parameters, with the only difference that "Transformer small" has no MASS pre-training, "LightPAFF" is our method applied on "MASS small".

**Inference Speedup of LightPAFF** As shown in Table 6, LightPAFF achieves nearly $5\times$ speedup over the big pre-trained model on both GPU and CPU, which demonstrates the advantages of LightPAFF for inference speedup.

| | #Param | Latency (GPU) | Speedup | Latency (CPU) | Speedup |
| --- | --- | --- | --- | --- | --- |
| MASS | 213M | 27 ms | 1.00 $\times$ | 94 ms | 1.00 $\times$ |
| LightPAFF | 42M | 6 ms | 4.50 $\times$ | 18 ms | 5.22 $\times$ |

Table 6: Inference speedup LightPAFF on MASS. Evaluation is conducted by generating one sentence at a time autoregressively. The GPU/CPU configurations follow that in last Section.

## 6 Further Analysis

To better understand LightPAFF, we conduct some deep analyses on LightPAFF in this section.

---

[5]https://github.com/mjpost/sacreBLEU

## 6.1 ABLATION STUDY

To study the impact of knowledge distillation in each stage, we conduct ablation studies by removing distillation from one of the stages. The results are shown in Table 7. We have several observations: 1) The comparison between Method 1 with 2 shows that without pre-training, using knowledge distillation in fine-tuning stage can improve the accuracy of the model. 2) Comparing Method 1 and 3 or Method 2 and 5, we see that using pre-training can always boost accuracy. 3) Comparing Method 3 and 4 or Method 5 and 6, it can be seen that using knowledge distillation in pre-training can boost the accuracy. 4) Comparing Method 3 and 5 or Method 4 and 6, it can be seen that using knowledge distillation in fine-tuning can boost the accuracy. Overall, the distillation in both stages are effective and necessary in LightPAFF.

| ID | PT | PT + KD | FT | FT + KD | BERT | GPT-2 | MASS |
|---|---|---|---|---|---|---|---|
| 1 (Transformer small) | | | ✓ | | 83.4 | 71.6 | 23.8 |
| 2 | | | | ✓ | 86.9 | 34.3 | 26.2 |
| 3 (Pre-trained small) | ✓ | | ✓ | | 89.7 | 25.5 | 28.4 |
| 4 | | ✓ | ✓ | | 90.8 | 21.6 | 29.5 |
| 5 | ✓ | | | ✓ | 92.0 | 22.1 | 31.0 |
| 6 (LightPAFF) | | ✓ | | ✓ | **92.9** | **18.8** | **32.2** |

Table 7: Ablation study on knowledge distillation in pre-training and fine-tuning stages, where "PT" means pre-training, "FT" means fine-tuning and "KD" means knowledge distillation. The results of BERT/GPT-2/MASS are the accuracy on SST-2 task, the perplexity on WikiText-2, and the BLEU score on WMT16 En-De (100K) respectively.

## 6.2 GENERALIZATION ANALYSIS

Previous works (Yang et al., 2018; Lan et al., 2018) have shown that knowledge distillation can help improve the generalization ability, and some studies (Shirish Keskar et al., 2016; Chaudhari et al., 2016) indicate the relationship between model generalization and the width of local minima in the loss surface, where models with wider local minima are more robust to perturbations. Inspired by Tan et al. (2019), we perturb the parameters of the models to observe the accuracy changes, in order to demonstrate that the knowledge distillation in both pre-training and fine-tuning stages in LightPAFF improves the generalization ability.

We perturb a model $\theta$ as $\theta_i^l(\sigma) = \theta_i^l + \bar{\theta}^l * \mathcal{N}(0, \sigma^2)$, where $\theta_i^l$ is the $i$-th parameter of $\theta^l$, where $\theta^l$ represents the parameter named by $l$ (e.g., $W_Q$, $W_K$ or $W_V$ in self-attention) in model $\theta$, and $\bar{\theta}^l$ is the average of parameter $\theta^l$. We sample from the normal distribution $\mathcal{N}$ with standard variance $\sigma$ where larger $\sigma$ represents bigger perturbation on the parameter. The results with varying $\sigma$ are shown in Figure 2. It can be seen that the accuracy of the model without knowledge distillation drops faster no matter in pre-training and fine-tuning stage, which demonstrates knowledge distillation in both pre-training and fine-tuning stages make the model more robust.

## 6.3 HYPERPARAMETER FOR DISTILLATION AND DIFFICULTY OF PRE-TRAINING TASKS

**Hyperparameter $\lambda$** LightPAFF employs a hyperparameter $\lambda$ (see Equation 1) to control the trade-off between knowledge distillation from the teacher and learning from the original label in the data. During pre-training, the optimal $\lambda$ according to our experiments is 1.0 for BERT, 0.7 for MASS and 0.4 for GPT-2. Bigger $\lambda$ means to use more information from teacher while less information from ground-truth label in the original data. This seems to suggest that the pre-trained BERT teacher model preserves the most information of the data, while the pre-trained GPT-2 teacher model preserves the least information of the data. To verify this, we further check the training results of the three models.

**Prediction Accuracy of Pre-training** For this study, we pre-train BERT, GPT-2 and MASS models on the same data corpus, with similar model configurations (12-layer BERT, 12-layer GPT-2

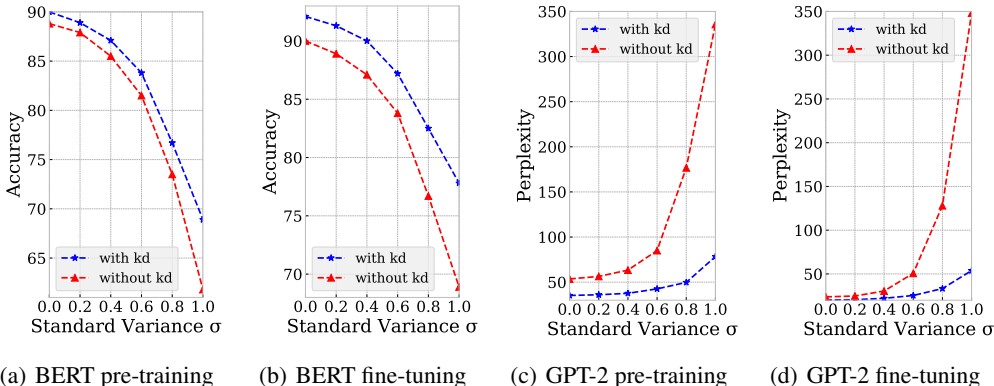

|  | (a) BERT pre-training | (b) BERT fine-tuning | (c) GPT-2 pre-training | (d) GPT-2 fine-tuning |

Figure 2: Generalization Analysis. The result of BERT is the accuracy of SST-2 on valid set while the result of GPT-2 is the perplexity of WikiText-2 on valid set.

| Model | Accuracy | Input Tokens |
|-------|----------|--------------|
| BERT  | 0.73     | 85%          |
| MASS  | 0.62     | 75%          |
| GPT-2 | 0.42     | 50%          |

Table 8: The comparison of accuracy, and average number of input tokens in the pre-training task between BERT, GPT-2 and MASS.

and 6/6-layer MASS, with hidden size of 768) and same vocabulary. We test the three models on a held-out dataset. In particular, we compute the accuracy of token predictions. The results are shown in Table 8. It can be seen that BERT achieves the highest accuracy while GPT-2 the lowest. Since the three models are of similar complexity (e.g., with similar architecture and similar number of parameters), our hypothesis is that the pre-training task of BERT is easiest and that of GPT-2 is the most difficult, which is verified intuitively as below.

**Task Difficulty of Pre-training**   Many pre-training tasks in natural language processing are to predict a token conditioned on a sequence of input tokens. BERT masks 15% tokens in a sequence and predicts these 15% tokens given the remaining 85% tokens[6]. GPT-2 pre-trains a language model, i.e., predicting every token in a sequence conditioned on all its preceding tokens. For a sentence with $n$ tokens, when predicting the $i$-th token, the model can see the previous $i-1$ tokens. Therefore, the average tokens that can be seen is $(n-1)/2$, which is about 50%. For MASS, 50% tokens can be seen in the encoder, while 50% tokens are predicted in the decoder side through language modeling, where on average 50%/2 tokens can be seen following the calculation in GPT-2. Therefore, to predict a token, on average MASS uses 75% tokens. To summarize, as shown in Table 8, to predict each token, on average BERT uses 85% tokens of a sequence, MASS uses 75% while GPT-2 only 50%. Thus, the difficulty of the pre-training tasks should be BERT<MASS<GPT-2, which is consistent with the observations in last two paragraphs.

## 7   CONCLUSION

In this work, we have proposed LightPAFF, a two-stage knowledge distillation method to reduce the size of the big pre-trained models, which transfers the knowledge from big pre-trained model into small model in both pre-training and fine-tuning stages. We formulate LightPAFF on BERT, GPT-2 and MASS models uniformly and conduct experiments on BERT, GPT-2 and MASS with 9 downstream tasks in total. The experimental results demonstrate that LightPAFF reduces the model size by nearly $5\times$ and improves the inference speed by $5\times \sim 7\times$, while achieves similar accuracy

---

[6]We ignore the 8:1:1 strategy in BERT, which does not affect the percentage much.

with the original BERT/GPT-2/MASS model. For future works, we will cover more pre-trained models and downstream tasks. On the other hand, we will also explore other methods such as quantization, pruning to compress the big pre-trained models.

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

APPENDIX

# 1 TWO-STAGE DISTILLATION FOR BERT, GPT-2 AND MASS

We formulate the knowledge distillation for pre-training and fine-tuning tasks in BERT, GPT-2 and MASS, where BERT (Devlin et al., 2018) pre-trains on masked language modeling[7] and fine-tunes on language understanding tasks, GPT-2 (Radford et al., 2019) pre-trains and fine-tunes both on causal language modeling tasks, while MASS (Song et al., 2019) pre-trains on masked sequence to sequence modeling and fine-tunes on sequence to sequence based language generation tasks.

## 1.1 KNOWLEDGE DISTILLATION IN PRE-TRAINING

**Masked Language Modeling**    The knowledge distillation loss for masked language modeling (Devlin et al., 2018) $\mathcal{L}_{MLM}$ is as follows:

$$\mathcal{L}_{MLM}(\theta) = -\sum_{x \in \mathcal{D}} \sum_{m \in \mathcal{M}} \sum_{k=1}^{|\mathcal{V}|} ((1-\lambda) \cdot \mathbb{1}\{x_m = k\} + \lambda \cdot Q(x_m = k|x^{\backslash \mathcal{M}}; \theta_T)) \cdot \log P(x_m = k|x^{\backslash \mathcal{M}}; \theta), \quad (2)$$

where $\mathcal{D}$ is the corpus for pre-training, $x$ is a sentence, $\mathcal{M}$ is the set of indices of the masked tokens in sentence $x$ and $x_m$ is the corresponding prediction of the masked token, $|\mathcal{V}|$ is the vocabulary size. $x^{\backslash \mathcal{M}}$ is the sentence with token indices in $\mathcal{M}$ are masked. The masking operation in BERT is as follows: 15% tokens are randomly masked, where 80% are masked with the token "[MASK]", 10% are masked with a random token, and the rest 10% are remained unchanged. $\mathbb{1}$ is the indicator function indicating if $x_m$ equals to $k$, $Q(x_m = k|x^{\backslash \mathcal{M}}; \theta_T)$ is the probability distributions for the predicted token generated by the teacher model $\theta_T$ which is fixed during training.

**Causal Language Modeling**    The knowledge distillation loss for causal language modeling (Radford et al., 2019) $\mathcal{L}_{CLM}$ is formulated as:

$$\mathcal{L}_{CLM}(\theta) = -\sum_{x \in \mathcal{D}} \sum_{m=1}^{|x|} \sum_{k=1}^{|\mathcal{V}|} ((1-\lambda) \cdot \mathbb{1}\{x_m = k\} + \lambda \cdot Q(x_m = k|x_{<m}; \theta_T)) \times \log P(x_m = k|x_{<m}; \theta), \quad (3)$$

where $|x|$ is the length of the sentence $x$, $x_{<m}$ is the tokens preceding position $m$.

**Masked Sequence to Sequence Modeling**    The knowledge distillation loss for masked sequence to sequence modeling (Song et al., 2019) $\mathcal{L}_{MSSM}$ is formulated as:

$$\mathcal{L}_{MSSM}(\theta) = -\sum_{x \in \mathcal{D}} \sum_{m=s}^{t} \sum_{k=1}^{|\mathcal{V}|} ((1-\lambda) \cdot \mathbb{1}\{x_m = k\} + \lambda \cdot Q(x_m = k|x_{<m}^{s:t}))$$
$$\times \log P(x_m = k|x_{<m}^{s:t}, x^{\backslash s:t}; \theta), \quad (4)$$

where $s$ and $t$ are the start and end position of the masked segment in masked sequence to sequence modeling, $x^{s:t}$ denotes the sentence segment from position $s$ to $t$, $x^{\backslash s:t}$ denotes the sentence where the segment from position $s$ and $t$ are masked. The masking strategy in Song et al. (2019) is as follows: the masked tokens in the sentence will be a $[\mathbb{M}]$ token 80% of the time, a random token 10% of the time and a unchanged token 10% of the time.

## 1.2 KNOWLEDGE DISTILLATION IN FINE-TUNING

**Language Understanding**    Language understanding task usually refers to the tasks like QQP (Iyer et al., 2017), SST-2 (Socher et al., 2013), MNLI (Williams et al., 2018), which requires understanding of the sentence in order to make predictions. The loss function of the knowledge distillation for fine-tuning on the language understanding task is formulated as:

$$\mathcal{L}_{LU}(\theta') = -\sum_{(x,y) \in \mathcal{D}'} \sum_{k=1}^{|\mathcal{V}|} ((1-\lambda) \cdot \mathbb{1}\{y = k\} + \lambda \cdot Q(y = k|x; \theta_T')) \times \log P(y = k|x; \theta'), \quad (5)$$

---

[7]BERT leverages masked language modeling and next sentence prediction for pre-training. Considering some works (Yang et al., 2019b; Liu et al., 2019; Joshi et al., 2019) have achieved good results without next sentence prediction, we only distill BERT on the masked language modeling task in pre-training.

where $\mathcal{D}'$ is the supervised training corpus in downstream tasks, $x$ is the input sentence, $y$ is the label and $|\mathcal{V}|$ is the size of label set, e.g., $y$ represents the sentiment classes for sentiment classification task, and $|\mathcal{V}|$ is the number of classes[8]. $Q$ is output probability of the teacher model $\theta'_T$ that is fine-tuned from pre-trained teacher model $\theta_T$, while $P$ is the output probability of student model $\theta'$ that is initialized from the pre-trained model $\theta$ and only differs from $\theta$ in the task-specific output layer.

**Language Modeling/Generation**  Language modeling/generation refers to the tasks like WikiText-2 (Merity et al., 2017), PTB (Mikolov et al., 2010) and WikiText103 (Merity et al., 2017). The knowledge distillation loss is

$$\mathcal{L}_{LM}(\theta') = -\sum_{x \in \mathcal{D}'} \sum_{m=1}^{|x|} \sum_{k=1}^{|\mathcal{V}|} ((1-\lambda) \cdot \mathbb{1}\{x_m = k\} + \lambda \cdot Q(x_m = k | x; \theta'_T)) \times \log P(x_m = k | x_{<m}; \theta'), \quad (6)$$

which is similar to that in Equation 3, except for that training corpus is from the specific downstream task, and teacher model $\theta'_T$ and student model $\theta'$ are already pre-trained.

**Sequence to Sequence based Language Generation**  Sequence to sequence based language generation tasks include neural machine translation, text summarization, grammatical error correction, text style transfer, etc. The knowledge distillation loss is formulated as

$$\mathcal{L}_{SS}(\theta') = -\sum_{(x,y) \in \mathcal{D}'} \sum_{m=1}^{|y|} \sum_{k=1}^{|\mathcal{V}|} ((1-\lambda) \cdot \mathbb{1}\{y_m = k\} + \lambda \cdot Q(y_m = k | y_{<m}, x; \theta'_T)) \\ \times \log P(y_m = k | y_{<m}, x; \theta'), \quad (7)$$

where $(x, y)$ is the supervised sentence pair, $|y|$ is the length of the target sentence.

## 2  DATASET

**BERT**  We select 3 different datasets to evaluate our method on language understanding tasks for BERT. SST-2 (Stanford Sentiment Treebank 2) is a movie review dataset with binary labels indicating positive or negative sentiment of the sentences. QQP (Quora Question Pairs) contains question pairs with binary labels indicating whether they are duplicate. More details about the above two datasets can be found in Devlin et al. (2018). PolyDis (Polyphone Disambiguation) consists of sentences with polyphonic characters whose labels are their pronunciations on Chinese Mandarin. PolyDis dataset contains 166,185 sentences for training and 43,426 sentences for testing with 79 most frequent polyphonic characters. To further reduce the gap between teacher model and lightPAFF, we add additional unlabeled source data in the fine-tuning stage of BERT distillation. For PolyDis, we simply extract unlabeled data which contains polyphonic words from the pre-training corpus. For SST-2 and QQP, we get unlabeled data of movie reviews and question pairs from Kaggle, a competition website with a lot of open datasets, and we remove all the data that occurs in the valid and test set of SST-2 and QQP. The additional dataset of SST-2 can be got from `https://www.kaggle.com/utathya/imdb-review-dataset` and QQP from `https://www.kaggle.com/develina/quora-pairs`

**GPT-2**  Different from BERT, GPT-2 (Radford et al., 2019) proposed a web scrape to filter data from web pages by emphasizing document quality. These resulting dataset, named as WebText, contains nearly 40GB text after de-duplication and basic cleaning operations. The size of WebText dataset is 2.5 times to BERT training corpus. Due to computation resource constraint, we do not use the huge training data as in GPT-2, but we believe using more data can achieve better accuracy. For language model tasks (PTB, Wikitext-2, Wikitext-103), each task has been split into train/valid/test, and the detail of each dataset is shown in Table 9:

---

[8]We make two clarifications here: 1) Language understanding can also take multiple sentences as input, such as QQP, where the two sentences are concatenated together and regarded as one sentence. Therefore, we just formulate the input as one sentence for simplicity. 2) Language understanding tasks can also take multiple tokens or labels as output, such as sequence labeling task like NER, polyphone disambiguation. As there is no explicit dependency when predicting different labels in BERT model, we just formulate the task as single label for simplicity.

|  | PTB | | | Wikitext-2 | | | Wikitext-103 | | |
|---|---|---|---|---|---|---|---|---|---|
|  | Train | Valid | Test | Train | Valid | Test | Train | Valid | Test |
| Tokens | 887K | 70K | 78K | 2.08M | 217K | 245K | 103M | 217K | 245K |

Table 9: Statistics of the three datasets of language modeling tasks in GPT-2.

**MASS**   We conduct experiments on one rich-resource and two low-resource machine translation datasets to evaluate the effectiveness of LightPAFF in sequence to sequence tasks. For rich-resource task, we use WMT17 Chinese-English (Zh-En). For low-resource scenario, We build the datasets by randomly sampling 100K bilingual data from the standard WMT14 English-French (En-Fr) and WMT16 English-German (En-De) datasets. For similar language pairs (i.e., En-Fr and En-De), we learn a jointed dictionary with 32,000 tokens. For discrete language pair (i.e., Zh-En), we respectively learn a dictionary for each language with 40,000 tokens.

## 3   TRADE-OFF BETWEEN ACCURACY AND MODEL SIZE

We compare the accuracy of our LightPAFF models with different model size, and report the results of PolyDis task on BERT, as shown in Table 10. It can be seen that there is a trade-off between accuracy and model size. Different models can be chosen for online deployment when taking accuracy, memory cost and inference latency into consideration.

|  | #Param | Acc. |
|---|---|---|
| BERT | 110M | 95.9 |
| LightPAFF | 30M | 95.5 |
| LightPAFF | 20M | 95.4 |
| LightPAFF | 8M | 93.9 |

Table 10: The accuracy of LightPAFF with different number of parameters on PolyDis task. We also list the results of the original BERT model with 110M parameters.

## 4   ANALYSIS ON THE USE OF UNLABELED DATA

During the fine-tuning in BERT, we leverage unlabeled data for knowledge distillation to reduce the accuracy gap between the teacher model and student model. We study how unlabeled data can help improve the accuracy of student model, as shown in Table 11. It can be seen that using unlabeled data for knowledge distillation can indeed boost the accuracy of the student model (LightPAFF).

| unlabeled data | SST-2 Acc. | QQP F1/Acc. | PolyDis Acc. |
|---|---|---|---|
| without | 91.3 | 67.5/87.7 | 94.5 |
| with | 92.9 | 70.6/88.6 | 95.4 |

Table 11: The accuracy comparison of LightPAFF without and with unlabeled data on the BERT fine-tuning tasks.

To further check if teacher model can benefit from unlabeled data, we also use BERT teacher model to generate labels of unlabeled data and then teach itself. The result on SST-2 test set shows that the accuracy drops 0.1% (from 93.5% to 93.4%), which means unlabeled data may not help the teacher model itself but help student models.

