# OpenReview forum: "LightPAFF: A Two-Stage Distillation Framework for Pre-training and Fine-tuning"
_ICLR.cc/2020/Conference — Reject_

### Official Review · AnonReviewer1 · 2019-10-18
**Official Blind Review #1**

**Rating:** 6

**Review:**

The authors propose a method for distilling language models such as BERT into much smaller versions with similar accuracy. The approach, therefore, saves both memory and computation resources.

The difference to existing methods is that the authors method also performs distillation during pretraining. The authors apply their method to BERT and run experiments, comparing the distilled BERT model with the original one as well as existing distillation methods.

The experiments are extensive with impressive results. I'm torn. On the one hand, these results show that this method (which is simple and straight-forward) reduces memory and computation time (latency) substantially while maintaining similar accuracy. It can be very useful in practice. On the other hand, the idea is so simple and straight-forward, that I'm not sure ICLR (a top machine learning conference) is the right place for it. I tend to think that the paper should get accepted though since the empirical work is really exhaustive and impressive.

**Experience Assessment:**

I do not know much about this area.

**Review Assessment: Checking Correctness Of Derivations And Theory:**

I assessed the sensibility of the derivations and theory.

**Review Assessment: Checking Correctness Of Experiments:**

I assessed the sensibility of the experiments.

**Review Assessment: Thoroughness In Paper Reading:**

I read the paper at least twice and used my best judgement in assessing the paper.

---

> ### Author Response · Authors · 2019-11-14
> **Response to Review #1**
>
> Thanks for your valuable comments and suggestions!
>
> Our response has been merged together here: https://openreview.net/forum?id=B1xv9pEKDS&noteId=S1lA8Ti9iS

---

### Official Review · AnonReviewer2 · 2019-10-23
**Official Blind Review #2**

**Rating:** 3

**Review:**


Summary: This work leverages knowledge distillation both in pre-training and fine-tuning stages to learn a more compact student model that approximates the performance of a teacher model. Extensive experiments with different knowledge distillation loss functions are conducted on a number of representative language representation models including BERT, GPT-2 and MASS to show the benefit of the method.

Strengths:

The numerical results show some promise of the proposed method to achieve competitive performance to the teacher network and outperform the considered baselines. The evaluated 9 downstream tasks cover a wide range of language understanding and generation problems and they do prove the effectiveness of knowledge distillation in boosting the accuracy.

Weaknesses:

My main concern is about the limited novelty of approach. The knowledge distillation method developed in the paper seems not particularly novel in principle. Although extensive numerical results are provided to show the effectiveness of knowledge distillation, the value-added beyond applying different distillation loss functions in different NLP tasks is still insufficient. Particularly, the loss function that combines maximal likelihood loss with KL divergence looks similar to those used in the paper [1]. The connection and difference to that related work need to be clarified.

[1] Yu, Dong, et al. "KL-divergence regularized deep neural network adaptation for improved large vocabulary speech recognition." 2013 IEEE International Conference on Acoustics, Speech and Signal Processing. IEEE, 2013.

=== update after author response ===

Thank you for the response. Although interesting and showing some promise in a bunch of applications, I am still not convinced that the proposed approach is novel enough in principle.  I thus stick to my assessment of this paper as borderline leaning to rejection.


**Experience Assessment:**

I have published one or two papers in this area.

**Review Assessment: Checking Correctness Of Derivations And Theory:**

N/A

**Review Assessment: Checking Correctness Of Experiments:**

I carefully checked the experiments.

**Review Assessment: Thoroughness In Paper Reading:**

I read the paper at least twice and used my best judgement in assessing the paper.

---

> ### Author Response · Authors · 2019-11-14
> **Response to Review #2**
>
> Thanks for your valuable comments and suggestions!
>
> The common response has been merged together here: https://openreview.net/forum?id=B1xv9pEKDS&noteId=S1lA8Ti9iS
>
> Thanks for pointing out the reference! We have cited it in the new version submitted today. While the loss functions in [1] and our work are similar, the methods and scenarios are different. [1] aims to regularize the adapted model not to stray too far from the original model, which is mainly used to combat overfitting when the adaptation data is small. While the loss in our method is used to transfer the knowledge from the big teacher model to the small student model in the pre-training and fine-tuning stage, respectively.
>
> [1] Yu, Dong, et al. "KL-divergence regularized deep neural network adaptation for improved large vocabulary speech recognition." 2013 IEEE International Conference on Acoustics, Speech and Signal Processing. IEEE, 2013.

---

### Official Review · AnonReviewer3 · 2019-10-23
**Official Blind Review #3**

**Rating:** 6

**Review:**

The paper sets to solve the problem that many language models such as GPT-2 despite having achieved great success in their language tasks are too big to deploy in applications. They propose LightPAFF, a framework that allows to transfer knowledge from a big teacher model to a lightweight student model, thus solving the deployability issue at hand. They conduct experiments showing LightPAFF achieves a 5x reduction in the number of parameters and a 5x-7x improvement on the inference speed, while roughly preserving performance.

The distillation framework the authors use in their methods is not new. It has been proposed in previous work, as the authors noted. As opposed to previous works, where distillation is performed in the fine tuning stage, the authors of LightPAFF propose a two stage distillation procedure instead that performs distillation at the pre training phase and fine tunes this distilled model for use in a downstream task using a big fine tuned teacher model and the dataset of the task. The experimental results show these results to be meaningful. They achieve better accuracy and similar compression than previous approaches. It's main weakness is that the method doesn't seem to be particularly new.

**Experience Assessment:**

I do not know much about this area.

**Review Assessment: Checking Correctness Of Derivations And Theory:**

N/A

**Review Assessment: Checking Correctness Of Experiments:**

I assessed the sensibility of the experiments.

**Review Assessment: Thoroughness In Paper Reading:**

N/A

---

> ### Author Response · Authors · 2019-11-14
> **Response to Review #3**
>
> Thanks for your valuable comments and suggestions!
>
> Our response has been merged together here: https://openreview.net/forum?id=B1xv9pEKDS&noteId=S1lA8Ti9iS

---

### Author Response · Authors · 2019-11-14
**Response to Review #1/#2/#3**

Thanks for your valuable comments and suggestions!

Although our method is conceptually simple, it has the following unique values to both the research community and industry applications:

1. It is the first time that knowledge distillation is introduced in the pre-training stage, while previous works mainly use knowledge distillation in the training of downstream tasks.

2. Our proposed framework is very general. It can support a variety of pre-training methods (masked language modeling (e.g., BERT, RoBERTa), causal language modeling (e.g., GPT/GPT-2), masked sequence to sequence modeling (e.g., MASS)) as well as downstream tasks (language understanding (e.g., GLUE tasks), language modeling (e.g., PTB, WikiText-2), sequence to sequence generation(e.g., neural machine translation, text summarization)). In this paper, we conduct comprehensive studies on three representative pre-training models (BERT, GPT-2, and MASS) and their corresponding downstream tasks (language understanding, language modeling, and sequence to sequence generation), with 9 tasks in total, and verify the effectiveness of the framework. There are contemporary works mainly focusing on compressing BERT for language understanding tasks, while we cover diverse pre-trained models and downstream tasks.
Our LightPAFF with two-stage knowledge distillation is very practical to solve real-world applications. LightPAFF has greatly reduced the online inference time (almost without loss of accuracy) compared with the original big pre-trained model. Actually it has already been served online in our internal production system. We will release our codes and models once accepted.

3. We also conduct deep analyses on LightPAFF, including 1) the ablation study to show the effectiveness of the distillation in both pre-training and fine-tuning stages; 2) we study why distillation can help the model in the pre-training and fine-tuning stages by generation analysis; 3) more interestingly, we also analyze how different pre-training methods (BERT/GPT-2/MASS) influence the behavior of knowledge distillation (see Section 6.3), by analyzing the connections between the difficulty and prediction accuracy among different pre-training tasks, as well as the weight tradeoff between the knowledge distillation loss and ground-truth label loss.

Although the basic idea of knowledge distillation is not new, the simplicity, general applicability (supporting many pretraining models and downstream tasks), and practical effectiveness (e.g., shipped online in our commercial products) of our framework bring solid contributions to the community and make LightPAFF well suited for ICLR 2020. Thanks again for your valuable comments and further considerations!

---

### Decision · Program_Chairs · 2019-12-19

**Decision:**

Reject

**Comment:**

This paper proposes a two-stage distillation from pretrained language models, where the knowledge distillation happens in both the pre-training and the fine-tune stages.  Experiments show improvement on BERT, GPT and MASS.  All reviewers pointed that the novelty of the work is very limited.